# Women Acute Myocardial Infarction—Identifying and Understanding the Gender Gap (WAMy-GAP): A Study Protocol

**DOI:** 10.3390/healthcare12100972

**Published:** 2024-05-09

**Authors:** Vincenza Giordano, Assunta Guillari, Vincenza Sansone, Maria Catone, Teresa Rea

**Affiliations:** 1Department of Biomedicine and Prevention, University of Rome Tor Vergata, 00133 Rome, Italy; vincenza.giordano@alumni.uniroma2.eu; 2Public Health Department, Federico II University Hospital, 80131 Naples, Italy; maria.catone@studenti.unina.it (M.C.); teresa.rea@unina.it (T.R.); 3Department of Experimental Medicine, University of Campania “Luigi Vanvitelli”, 81100 Naples, Italy; vincenza.sansone@unicampania.it

**Keywords:** symptoms, myocardial infarction, women, delay in seeking treatment

## Abstract

Barriers to accessing care and misinterpretations of ischemic heart disease symptoms due to lack of awareness contribute to women’s delay in seeking care. Women may delay seeking treatment for up to 3 h or even up to 5 days. They often perceive themselves to be at low risk of cardiovascular disease (CVD) and prioritize family responsibilities or household chores. The causes of this delay are multifactorial and influence the decision-making process, particularly in the pre-hospital phase. The objective of this study protocol is to evaluate prodromal symptoms and identify risk behaviors in women with acute myocardial infarction (AMI). This is a protocol for a multicenter study that will be conducted using the mixed-method methodology using the McSweeney Acute and Prodromal Myocardial Infarction Symptom Survey (MAPMISS) to evaluate symptoms and semi-structured interviews to investigate behaviors. This study protocol is intended to fill an important knowledge gap on premonitory and acute symptoms of AMI in women in Italy, as well as to understand the causes and mechanisms underlying delays in accessing healthcare services during an acute event such as AMI. The investigation of this issue will facilitate the removal of gender-related inequalities in the diagnosis and treatment of acute myocardial infarction while also fostering dialogue on the barriers to behavior change.

## 1. Introduction

Cardiovascular diseases (CVD) account for 35% of all deaths in women, yet their impact is not thoroughly understood by the female population [1]. Understanding of the pathophysiological mechanisms and manifestations of ischemic heart disease in women is still limited, and the use of male-type diagnostic criteria contributes to delaying or compromising diagnosis. The literature shows an underrepresentation of women in clinical studies [1]. Therefore, often underestimated factors that may be influenced by sex and interaction with women’s social and physical environment (such as socio-economic deprivation, depression, anxiety, and low health literacy) need to be considered [1,2]. Ischemic heart disease was the leading cause of CVD mortality among women worldwide in 2019. In particular, within the spectrum of ischemic heart disease, conditions such as myocardial infarction with nonobstructive coronary artery disease (MINOCA) and ischemia with nonobstructive coronary artery disease (INOCA) increased [1,3]. Research highlights that women are 50% more likely than men to be under-diagnosed after an acute myocardial infarction (AMI), reducing their chances of receiving timely life-saving interventions or recommended heart attack treatment [4,5]. Women frequently exhibit alternative presentations of acute coronary syndrome (ACS), such as spontaneous coronary artery dissection and vasospasm, in addition to the plaque rupture commonly observed in men [6]. Moreover, across the spectrum of ACS, women tend to receive less invasive procedures and are prescribed fewer pharmacological interventions than men [6]. The early diagnosis and management of cardiovascular risk factors are crucial to improving women’s health and reducing early mortality. Although known factors such as hypertension, dyslipidemia, diabetes, obesity, unhealthy diet, sedentary lifestyle, and smoking are important contributors to heart disease, there are other less recognized factors, such as psychological, social, economic, and cultural factors, that significantly influence cardiovascular disease in women [1]. Depression, domestic violence, socio-economic status, and social and cultural roles have a significant impact on women and are increasingly recognized as key factors in the development and manifestation of these diseases. In addition, conditions related to reproductive health, such as gestational hypertension and diabetes, early menopause, and polycystic ovary syndrome, may further increase the risk of cardiovascular disease in women [1]. Barriers to accessing care, such as fear [6], underestimation of symptoms, and perceived low risk of heart disease, frequently lead women to delay from one hour to several days before seeking medical attention. This delay is due to several factors, including attempts to manage symptoms independently, prioritization of social and family responsibilities, self-medication, and the time taken to involve the most appropriate family member in decision-making [7,8,9,10,11,12,13,14]. Factors such as older age, low education, and socioeconomic status, along with lower awareness of cardiac risk, contribute to the delay in accessing health services [15,16]. Furthermore, women frequently assume a caregiver role [17], which can result in hesitation to seek urgent medical care, consequently reducing emergency room visits. However, help-seeking behavior is influenced by social factors, as women try to maintain personal and social integrity and continue normal activities until symptoms become unbearable [17,18,19]. Many women feel relieved when the decision to seek medical care is made by a family member, and prefer to wait for the most appropriate time for their partner to accompany them if symptoms are stable, described by a woman as “*symptoms not requiring a third nitroglycerin tablet*”) [14], instead of calling the territorial emergency service. However, there is an additional delay when preparing to go to the hospital, during which women feel compelled to complete various tasks such as showering, changing clothes, and preparing necessary items for the hospital visit. Even when symptoms worsen during these activities, women tend not to interrupt them, but adopt faster strategies to complete them, ensuring that everything is done without leaving anything unfinished [14]. This behavior extends to women in full-time employment, as they seek permission to leave work or arrange for someone to cover their duties before heading to the emergency room [14]. To date, few studies [7,20,21] have examined the psychological and social processes contributing to prehospital delays in women with acute myocardial infarction. Women with myocardial infarction often experience associated intermittent symptoms that may last for hours, days, or weeks [13,14,22,23,24], such as shortness of breath, palpitations, nausea, or pain radiating to the arm, neck, between the shoulder blades, and jaw. Because of the variety and intermittency of symptoms, both patients and healthcare professionals may underestimate the clinical signs of myocardial infarction in women [24]. 

When individuals seek medical attention before hospitalization for acute events like cardiac ischemia, the process typically involves three distinct phases [7]: (1) the time between the onset of symptoms and the decision to seek medical care, (2) the time between this decision and contact with medical personnel, and (3) the time spent between the first medical contact and arrival at the hospital. Particularly, women’s experience of myocardial infarction symptoms may influence their healthcare-seeking behavior before hospitalization. As the initial stage of pursuing medical treatment, which revolves around seeking care before admission, depends entirely on the decision-making process of the woman experiencing symptoms [25], numerous theoretical models have been devised to investigate behaviors during illness: in the health belief model, according to which if a person experiences symptoms of ischemia, such as those linked with a heart attack, the decision to consult a doctor is influenced by the perceived severity of symptoms and the perceived benefits or utility of seeking medical attention to address the situation [26]; in the self-regulatory model of cognition and behavior regarding illness, the individual’s representation of illness is shaped by previous experiences of illness, known as episodic memory [27,28,29,30]; in symbolic or role-based interactionism theory, reducing delays in seeking medical care for acute myocardial infarction hinges on the patient’s capacity to accurately recognize the symptoms and signs of the attack. This recognition is further reinforced by the support offered by individuals in the patient’s environment, who validate the accuracy of this identification [31]. The most popular model for explaining how women cope with the symptoms of acute myocardial infarction is the Leventhal model (Figure A1), known as the disease self-regulation model [29,30]. This model considers *Health Beliefs*, past experiences of illness, and external influences such as culture and social support [32,33,34]. Leventhal emphasizes the importance of examining people’s beliefs and habits in health management rather than focusing on their personality, suggesting that illness experiences influence health perception and behavior [27,28,29,30]. Finally, Leventhal’s model is useful for understanding how culture influences healthcare-seeking, considering that individual responses can be modulated by the cultural environment in which one lives [35].

Although there have been improvements in awareness regarding heart health, there is still a common wrong idea in our society: the false belief that heart attacks are primarily a men’s problem. This entrenched belief leads many women to not adequately consider their risk of heart disease, with potentially serious consequences for their health [36]. Furthermore, owing to the atypical presentation and prodromal symptoms not being recognized promptly, women are less likely to be admitted to intensive care, with a consequent increase in in-hospital mortality [37]. 

This study is intended to fill an important knowledge gap regarding premonitory and acute symptoms of AMI in women in Italy, as well as to understand the causes and mechanisms behind delays in access to healthcare services during an acute event such as AMI. Focusing exclusively on women, the study will prioritize sex-specific research on heart disease in women and intervention strategies, following the recommendations of the Lancet Commission [1]. By identifying premonitory and acute symptoms of AMI and the decision-making patterns women make to go to the hospital, the study will be able to inform and guide targeted educational interventions dedicated to healthcare providers and patients themselves. This will be critical in improving the prompt recognition of symptoms and early detection of heart disease in women, thus contributing to the prevention and optimal management of these conditions. In Italy, there are no exhaustive studies that analyze how the social context influences women’s expectations, interpretations, and actions, and consequently, the decision-making process underlying access to the healthcare system. Nurses are in a privileged position to protect and educate patients so that they recognize the signs of AMI, encouraging them to pay attention and trust signals from their bodies. Women should be encouraged to prioritize health and not postpone medical visits for family or work reasons. It is necessary to investigate to better understand the impact of sex-specific, psychosocial, and socioeconomic risk factors on ischemic heart disease in women and to evaluate the effectiveness of intervention strategies.

The primary objectives of this study include:(a)To investigate health-seeking behaviors and pre-hospital management by women with ischemic heart disease.(b)To identify psychosocial and socioeconomic factors that may influence delays in accessing healthcare services by women with ischemic heart disease.(c)To analyze the relationships between sex-specific risk factors, psychosocial and socioeconomic factors, and delays in accessing healthcare services by women with ischemic heart disease.

The secondary objectives are:(a)To evaluate prodromal cardiac symptoms in women with ischemic heart disease and compare them with those in the healthy population.(b)Validation of the *McSweeney Acute and Prodromal Myocardial Infarction Symptom Survey* (MAPMISS) questionnaire to evaluate the risk of ischemic heart disease in women in the short term (3 months).

## 2. Materials and Methods

### 2.1. Study Design

A multicenter study will be conducted using a mixed-method methodology (quantitative and qualitative) with a convergent parallel design. This will allow us to obtain information on variables that cannot be completely understood through quantitative investigations alone.

### 2.2. Samples and Setting

The quantitative part of the study will involve a non-probabilistic convenience sample of women with ischemic heart disease divided into three groups:✓Group I: patients diagnosed with ischemic heart disease confirmed by a cardiologist in the cardiology intensive care unit (ICU)/cardiology unit without a previous history of ischemic heart disease;✓Group II: patients diagnosed with ischemic heart disease with a previous history of ischemic heart disease confirmed by a cardiologist in ICU/cardiology units;✓Group III: Women without ischemic heart disease who attended cardiology clinics for a cardiac check-up and were diagnosed as healthy after cardiac evaluation by a cardiologist (healthy group).

The study will be conducted in ICU and cardiology units (Group I and Group II), as well as in outpatient cardiology clinics (Group III) of public, private, and convention facilities within the Campania Region. We will proceed with data collection once authorization has been obtained from each facility, and participants will provide consent to participate. Data collection will occur at two time points: initially at Time 0 for all three groups and then 3 months later for Group III to assess the potential onset of ischemic heart disease.

### 2.3. Sample Size 

The sample size was determined by considering an expected prevalence of 2.1% and calculated with a confidence level of 95%, with an acceptable standard error of 5% (0.05). The sample will therefore be made up of approximately 382 women (Groups I and II) to guarantee a minimum sample size with a confidence interval of 95%. Regarding Group III (healthy women), since we lack data on the number of women in Campania undergoing preventive cardiological outpatient checks, we will proceed with an estimate based on a sampling of women from the cardiology clinics. Random sampling, stratified by age (ranging from 20 to 85 years), will be conducted to ensure the selection of a representative sample of patients. For the qualitative phenomenological investigation, women who meet the inclusion criteria will be interviewed to gather in-depth information on their experiences, with particular attention to the specific context. Therefore, the sample size will be determined by data saturation and the emergence of sufficient themes, meaning that recruitment for interviews will continue until new interviews yield additional significant information or reveal emerging themes, as documented in the literature [38,39,40].

### 2.4. Criteria for Eligibility 

The quantitative study sample will be obtained according to the following criteria (Table 1):

The qualitative study sample will be obtained according to the following criteria (Table 2):

### 2.5. Study Measurement Tools

Several tools will be used to investigate myocardial infarction symptoms and risk behaviors that lead to delays in accessing care.

(*) *Questionnaire for socio-demographic variables*: date of birth, country of origin or birth, nationality, duration of residence in Italy, marital status, educational level, work activity, family income, and presence of other people who support the income (*ADDITIONAL SHEET*).

(*) *McSweeney Acute and Prodromal Myocardial Infarction Symptom Survey* (MAPMISS) [41,42,43,44,45]: the MAPMISS is among the few tools designed to capture the presentation of acute and prodromal symptoms of coronary heart disease and myocardial infarction in women. It has three sections:

*(1) “acute symptoms”*: administered following a cardiac event (e.g., myocardial infarction);

*(2) “prodromal symptoms (PS)” section*: 30 questions to be administered in the absence of a known cardiac event. Prodromal symptoms (PS) are defined as symptoms that (1) are new or increase in intensity or frequency before AMI, (2) are intermittent before MI, and (3) disappear or return to previous levels after AMI [44]. They are of particular interest because this section can be used for screening during a healthcare encounter and therefore should facilitate early diagnosis and treatment. Women rate each of the reported PS based on intensity (mild, medium, or severe), frequency, and time of onset (within the last month, two months, or three months). A prodromal symptom score is then calculated based on the product of the intensity and frequency of the symptoms (range: 0–21). Finally, the overall PS score is calculated by summing the individual symptom scores (range: 0–630).

*(3) background section*: Demographic characteristics, comorbidity conditions, other risk factors, and current medications.

The MAPMISS was developed through an analysis of symptoms reported by women of different age groups, ethnicities, and races, and employs their most common language expressions to describe the experienced symptoms [42,43,45]. Initial psychometric testing indicated that the MAPMISS has high content validity and acceptable test–retest reliability for women with known coronary heart disease [42]. The reliability of the MAPMISS as a screening tool has been established in Caucasian and African-American women (test–retest reliability r = 0.92, *p* < 0.001; concordance correlation coefficient = 0.92, *p* < 0.001) [42,43]. The content validity of the MAPMISS was assessed by seven content experts during the development of the instrument [42]. In our study, the MAPMISS will undergo a validation process within the Italian context. 

(*) *Mini-Cog* [46]: It evaluates cognitive impairment and combines two simple cognitive tasks (three-item word memory and clock drawing) with an empirical algorithm for scoring. A total score of 0, 1, or 2 indicates a higher likelihood of clinically important cognitive impairment. A total score of 3, 4, or 5 indicates a lower likelihood of dementia but does not rule out some degree of cognitive impairment.

(*) *Semi-structured interview*: Interviews will last a maximum of 45 min, and will include open-ended questions regarding (1) the patient’s experiences from the onset of symptoms to the decision to seek medical attention; (2) questions about patients’ physical and emotional feelings during the experience, decision-making processes (e.g., who they talked to and who they asked for advice), and beliefs about what was happening at the onset of myocardial infarction symptoms. A matrix of structured interview questions was developed specifically for this study based on a literature review that examined the concepts and factors associated with women’s delay in seeking medical care [24,47,48]. The interviews will be conducted during the patient’s hospitalization or 2–3 months after discharge, as in the study by Blakeman et al. [49].

### 2.6. Validity and Reliability/Rigor

In this study, we will use the MAPMISS instrument, which has not previously been validated in Italian for its psychometric properties. However, this instrument has a solid theoretical basis and has already been tested for validity and reliability in other languages [42,43].

### 2.7. Project Phases and Data Collection

To conduct this study (Figure A2), we will first proceed with the cross-cultural and linguistic validation of the MAPMISS, and subsequently, psychometric evaluation of the instruments will be conducted. For the validation of the MAPMISS tool, the author’s consent was requested upon receipt of the questionnaire. Following the validation of the MAPMISS tool, we intend to administer it to the study population. Concurrently, we will conduct a qualitative investigation using semi-structured interviews. This qualitative part aims to identify psychosocial and socio-economic factors influencing pre-hospital management by women with ischemic heart disease and their access to healthcare services. The investigation also seeks to explore the complex interplay between sex-specific, psychosocial, and socioeconomic risk factors that may contribute to delays in accessing healthcare services among women with ischemic heart disease.

### 2.8. MAPMISS Validation Process 

To conduct this study (Figure A2), we will first proceed with cross-cultural and linguistic validation of the MAPMISS. The process follows the phases of the model proposed by Beaton et al. [50], which are divided into the following phases: (1) forward translation, (2) synthesis, (3) backward translation, (4) expert committee review; and (5) pre-testing. 

✓*Forward translation, synthesis, and backward translation*: After obtaining permission from the authors for translation, cultural adaptation, and validation in Italian, forward translation will be carried out: two native Italian translators will independently translate the questionnaire, producing two versions (T1 and T2); then, the two translators will produce a common version (T-12) of the two previous translations (T1 and T2); the common version produced by the Italian translators will be translated backward, with two native English translators each producing one translation (BT1 and BT2) of the synthesis (T-12).✓*Expert committee review*: A fundamental step in this phase is content validation, which will be based on the evaluation of the expert committee (seven content experts, including cardiologists, cardiology nurses, and five women to validate the content of the instrument, as in the original study) [42]; thereafter, we will proceed with calculation of the CVR (*Content Validity Ratio*), I-CVI (*Content Validity Index* of the individual elements), S-CVI/Ave (average of the I-CVI of the elements of the entire instrument) indices, and S-CVI/UA (*Content Validity Index* with general agreement calculation). Where applicable, the process will continue until acceptable indices of content-related validity or equivalence are achieved to obtain a pre-definitive Italian version of the *McSweeney Acute and Prodromal Myocardial Infarction Symptom Survey* (MAPMISS).✓*Pre*-testing: A pilot study will be carried out with 10–15 women with acute myocardial infarction to test the comprehensibility of the instrument and the practical aspects of its administration. A qualitative analysis of the participants’ responses will be conducted to make any changes to the items; they will be asked to assess whether the language used in the translated version is understandable, whether the questions are adequate or need to be rephrased, and whether the translation can be culturally accepted to assess face validity.✓Subsequently, we will conduct a validation study on a larger sample of participants to conduct factor analysis. The purpose of this phase is to ensure that the questionnaire measures what it aims to measure by evaluating the coherence between the questions and the identified.

### 2.9. Method of Conducting the Qualitative Investigation

In the present study, we will adopt a qualitative approach based on the methodology outlined by Cohen et al. [51], which combines Husserl’s descriptive phenomenology with Gadamer’s interpretation. This method is particularly suitable for exploring complex and emotional experiences, focusing on daily activities, and avoiding prejudices and personal interpretations. We will ensure maximum confidentiality of participants, who will have the freedom to withdraw from the study at any time. Data collection will follow the stages defined by Cohen et al. [51], starting with “bracketing” to suspend prejudices, followed by interviews with open questions and notes on non-verbal language. The data collection will be deemed complete once no novel themes emerge from the interviews. The collected data will be transcribed and the identified themes will be confirmed or corrected by the participants, thus ensuring the validity of the results. This phenomenological methodology aims to provide an authentic and in-depth understanding of women’s experiences, thus contributing to our knowledge of the phenomenon under study. 

Sampling and interviewing will continue until saturation is reached or the extracted themes become redundant. The concept of saturation is widely used in qualitative research, where it is typically called “data saturation” or “thematic saturation” [39]. Saturation is “the assurance of qualitative rigor most often promoted by authors for reviewers and readers” [52]. In this study, the “code meaning” approach will be used, which does not focus on counting codes as a basis for determining saturation (as in other approaches), but rather on achieving a comprehensive understanding of codes as an indicator of saturation. It involves reviewing an interview and noting each problem (or code) identified [39]. In subsequent interviews, this approach aims to check if any new aspects, dimensions, or nuances of that code are uncovered until nothing new is identified, indicating it has reached saturation. Specifically, the method of Graneheim and Lundman [53] will be used for the analysis of emergent themes. This method includes the following steps: transcribing the entire interview immediately after the end of the interview, reading the entire text to gain a general understanding of its content, identifying the meaning units and extracting the primary codes, classifying similar codes into subcategories and complete categories (manifest content), and then determining the themes (the latent content). In this way, the first units of meaning will be identified in sentences or paragraphs in the context of the interviews, and will then be condensed. The condensed meaning units will be abstracted and labeled as primary codes. In primary coding, the text of each interview will be read several times, and its original sentences will be extracted and recorded [53]. To avoid potential bias associated with qualitative sampling, the criteria of Lincoln and Guba [54] will be used to ensure the scientific rigor of the present study. Continuous sampling up to data saturation will ensure credibility. For the present study, credibility will correspond to internal validity and will be the criterion for ensuring that the study examines what it was supposed to examine. Reliability will be guaranteed by the triangulation technique. Moreover, the involvement of two or more researchers in data analysis will provide a robust confirmation of the results from diverse perspectives. Confirmability of the findings will be ensured by a briefing of the participants and by their confirmation of the themes derived from the data collected during the interviews. Transferability will be assured by the participants’ comprehensive descriptions of their experiences, accompanied by a detailed depiction of their sociodemographic characteristics [54].

## 3. Data Analysis and Statistics

Data from the questionnaire will be analyzed using descriptive (e.g., average, standard deviation, and frequencies) and inferential statistical techniques. Statistical analysis will explore the existence of statistically significant relationships between the investigated variables (*p* < 0.05). Associations between variable categories will be assessed for significance using the χ^2^ test or *Fisher*’s exact test for regression logistics. The semi-structured interviews will be transcribed in full and analyzed according to *Cohen*’s phenomenological methodology [51]. If there are difficulties in extracting the main themes and sub-themes, we will use the qualitative statistics software NVIVO 14.

### Ethical Aspects

The study will be conducted in accordance with the Declaration of Helsinki and was approved by the Campania 3 Ethics Committee (protocol code no. 151) on 5 June 2023. 

## 4. Discussion

In patients with myocardial infarction, symptom recognition is essential for determining medical interventions or seeking help. However, women not only do not recognize the importance of prodromal symptoms [14,55,56,57] but often delay seeking treatment when infarct symptoms appear compared to men [7,14,56,57,58,59,60,61]. As early as 2001–2002, the *American Heart Association* and *the National Heart*, *Lung*, and *Blood Institute* introduced the “*Go Red for Women*” campaign and the “*Heart Truth*” program, both of which focused on young women aged 40–60 years to increase awareness of CVD and symptom recognition of myocardial infarction [61]. Despite the effectiveness of awareness programs in promoting women’s utilization of emergency services and enhancing symptom recognition, several studies agree that women still experience delays in seeking timely care [62,63,64]. These delays contribute to prolonged ischemic periods and underscore the ongoing challenges in ensuring prompt access to healthcare by female patients [65]. Although social and environmental issues have been well identified, the role of the female gender as a secondary and discriminatory factor in healthcare has been inadequately explained and often overlooked in existing literature, with few studies specifically addressing this issue [66]. Therefore, to better understand the complex decision-making process behind the delay in requesting assistance, it is fundamental to analyze the personal and individual perspectives of women who experience an acute event [24]. It is therefore necessary not only to integrate theoretical concepts with findings from the existing literature on pre-hospital delay [67] but also to adopt innovative research methodologies. This includes investigating the underlying psychosocial factors that contribute to pre-hospital delay in young women with myocardial infarction, beyond mere knowledge deficits, and exploring the interrelationships between these factors to identify actionable points for intervention [14,66].

The results of this study, which will explore sex-specific risk factors, as well as psychosocial and socioeconomic factors and their possible relationships with delays in accessing services by women with acute myocardial infarction (AMI) in Italy can be interpreted in the broader context of cardiovascular health disparities. In particular, these findings may confirm what was highlighted in the INTERHEART study [68], which analyzed the impact of psychosocial factors on the risk of myocardial infarction in a large sample of subjects from various countries. It has been demonstrated that variables such as occupational and domestic stress, financial problems, stressful life events, and depression are all factors associated with an increased risk of myocardial infarction. It is important to note that socioeconomic and psychosocial factors are often correlated and interact with each other, collectively influencing cardiovascular risk [68]. Therefore, understanding how these factors influence delays in accessing services can help delineate intervention strategies aimed at reducing health disparities. Additionally, the results will examine prodromal and acute symptoms of myocardial infarction, such as chest pain, typically experienced by men, which often does not have a significant prognostic value in women, who instead present with atypical and prodromal symptoms [69]. However, a major challenge for women is the variability in symptoms, which are often atypical and therefore less likely to manifest as chest pain [70]. The more frequent atypical symptomatology in women leads to attributing the observed symptoms to non-cardiac etiologies, which in turn are associated with treatment delays [71]. It has been observed that women present with symptoms that differ from those of men when myocardial infarction is suspected. These symptoms include pain in the upper back, arm, neck, and jaw, as well as unusual fatigue, respiratory difficulties, indigestion, nausea/vomiting, palpitations, weakness, and a sense of agitation [72]. Moreover, women tend to often experience prodromal symptoms, which are overlooked by both physicians and the women themselves [73]. These prodromal symptoms, such as unusual fatigue, sleep disturbances, and anxiety, can be early warning signs of an imminent myocardial infarction, indicating the early stages of heart disease [74]. It is crucial to consider gender differences in cardiac symptoms and ensure that women receive timely evaluation and treatment.

The results of the study will have a significant impact on modeling strategies for healthcare providers and policymakers. Women’s cardiac health involves a wide range of issues, including economic, legal, regulatory, psychosocial, ethical, religious, cultural, environmental, community, and policy aspects at both local and global levels. Identifying opportunities to improve symptom detection and the receipt of timely acute care is an important goal for healthcare providers, researchers, and policymakers to improve outcomes for women with AMI. Study findings regarding premonitory and acute symptoms of AMI, along with the causes of delays in care, could drive increased awareness campaigns for women on the importance of recognizing cardiac symptoms (increase in awareness) and seeking timely help as reported by the Lancet Report on Cardiovascular Disease in Women [1]. In addition, developing training programs for healthcare providers could prevent the underestimation of atypical symptoms reported by women, ensuring early diagnosis and better prevention of cardiovascular disease in this patient population. As the study also aims to identify risk factors that contribute to the overall burden of cardiovascular disease in women (including psychosocial and socioeconomic factors), it may facilitate the development of medical and community awareness initiatives to address them. The findings of the study may indicate a need for international health organizations to prioritize funding for cardiovascular disease prevention programs for women from socio-economically disadvantaged regions. To improve the prevention and treatment of cardiovascular diseases in women, it is essential to strengthen healthcare systems by involving healthcare providers. This requires an integrated approach that carefully considers the specificities of heart diseases in females. It is crucial to engage physicians, healthcare providers, and patients as partners in the detection and management of cardiovascular diseases in females worldwide [75]. Furthermore, it is important to implement platforms that allow patients to easily access screening and risk factor assessment, such as through pharmacies or digital technologies [76].

Limitations of the study include convenience sampling for both healthy and sick women for both methodologies (quantitative and qualitative). Another limitation is the hospital setting in which the interviews will be conducted, as it may compromise the accuracy of recalling events, as women may still be emotionally affected by the onset of the acute event.

Strengths, on the other hand, include validation in Italian and the use of a multidimensional instrument that has already been validated in several languages, such as English [41]. In addition, enrolling both women with acute myocardial infarction and women without a diagnosis of coronary artery disease who present with similar symptoms will allow us to assess whether women with acute myocardial infarction report significantly more symptoms than healthy women. This could also help determine whether prodromal symptoms are predictive of a coronary event. Finally, 3-month follow-up of healthy women is an essential strength to better understand the clinical significance of prodromal symptoms.

## 5. Conclusions and Expected Results

This study aims to validate the McSweeney Acute and Prodromal Myocardial Infarction Symptom Survey (MAPMISS) scale in Italy, which holds the potential to identify cardiac symptoms and experiences before myocardial infarction, thus providing a screening tool for women. The study will also identify psychosocial factors (such as stress, social support, anxiety, and depression) and socioeconomic factors (such as income, education level, and accessibility to health services) that may hinder timely access to care for women with ischemic heart disease and how some risky behaviors may influence the complex decision-making process in this specific patient population during an acute event. However, in addition to exploring the symptoms of myocardial infarction, it will be critical to understand and address the cognitive and emotional factors that influence decisions during a cardiac emergency event. Identifying these factors and enhancing their understanding among women hold the potential to lead to innovative educational and preventive interventions aimed at reducing delays in seeking care for myocardial infarction. This article describes a study protocol, hence the conclusions presented are probabilistic. We are confident that the results of the study will closely align with our conclusions. 

## 6. Relevance in Clinical Practice and Future Developments

We believe that the present study can bring several benefits to clinical practice.

Current knowledge does not adequately address the health problems associated with acute myocardial infarctions in women. Women experience different heart attack symptoms than men and are often diagnosed at a later stage. Understanding these differences is crucial for ensuring timely care. Therefore, a targeted investigation into this issue could help eliminate sex disparities in the diagnosis and treatment of heart attacks by improving understanding of the causes, risk factors, and specific preventive strategies, thus reducing the number of avoidable deaths. Furthermore, examining barriers to behavioral change, such as lack of time or financial resources, and providing support to women to identify solutions without judging their lifestyle choices could support their personal development and motivation. Understanding women’s perceptions of their surroundings could help reduce delays in accessing care, improve treatment outcomes, optimize the allocation of healthcare resources, and reduce the social and economic burdens associated with cardiovascular disease.

## Figures and Tables

**Table 1 healthcare-12-00972-t001:** Summary of quantitative study eligibility criteria.

Groups	Criteria for Eligibility
**Group I** (women with a definitive diagnosis of ischemic heart disease from a cardiologist)	Aged between 20 years and 85 years; No previous history of ischemic heart disease; Optimal and stable physical conditions for being interviewed; No cognitive impairment; Ability to speak and understand the Italian language.
**Group II** (women with a confirmed history of ischemic heart disease from a cardiologist)	Aged between 20 years and 85 years; At least one previous history of ischemic heart disease; Optimal and stable physical condition for being interviewed; No cognitive impairment; Ability to speak and understand the Italian language.
**Group III** (control group)	Aged between 20 years and 85 years; Healthy women in terms of cardiac health status (assessed by a cardiologist using physical examinations and diagnostic tests (electrocardiogram and/or echocardiogram and/or ergometric test); No incidence of ischemic heart disease for at least 3 months after the researchers have completed the questionnaire (follow-up); Possibility of follow-up for the occurrence of ischemic heart disease up to 3 months after sampling; No cognitive impairment; Ability to speak and understand the Italian language.

**Table 2 healthcare-12-00972-t002:** Summary of qualitative study eligibility criteria.

Group	Criteria of Eligibility
Women with a definitive diagnosis of ischemic heart disease from a cardiologist	Aged between 20 years and 85 years; Favorable and stable physical conditions for conducting interviews; Consent to be videotaped during interviews; Optimal and stable physical conditions for being interviewed; No cognitive impairment; Ability to speak and understand the Italian language.

## Data Availability

Data sharing is not applicable to this article.

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
