# Peer review of "Women Acute Myocardial Infarction—Identifying and Understanding the Gender Gap (WAMy-GAP): A Study Protocol"

_healthcare, 2024, doi:10.3390/healthcare12100972_

Round 1
Reviewer 1 Report
Comments and Suggestions for Authors
The study adopting both quantitative and qualitative data adds value to the study, by incorporating strategies to address uncertainties in population estimates and ensuring the adequacy of sample sizes for both types of analysis. However, it would be beneficial for the study to provide more detail on how data saturation will be assessed and how the emerging themes will be identified and analyzed.
The discussion is concise and summarizes the study's findings with existing literature, identifies the significance of the research problem, and outlines practical implications for clinical practice and future research. However, to strengthen the discussion further, it could consider addressing potential limitations of the study and offering more concrete recommendations for translating research findings into actionable strategies for healthcare providers and policymakers.
Furthermore, the connections between the study's findings and the broader context of cardiovascular health disparities should be highlighted so as to enhance the discussion's depth and relevance.
This is a very good protocol. The abstract is well written with well-defined objectives, methodology is clear.
Please check and edit the grammar.
Under section 3.2 the group classification needs to be corrected to Groups I-III
Potential limitations should be acknowledged.
How will the researchers address potential bias associated with qualitative sampling?
Comments on the Quality of English LanguageThe entire article needs to be reviewed for grammar and clarification of statements.
See line 79-81
See line 86
Author Response
-The study adopting both quantitative and qualitative data adds value to the study, by incorporating strategies to address uncertainties in population estimates and ensuring the adequacy of sample sizes for both types of analysis. However, it would be beneficial for the study to provide more detail on how data saturation will be assessed and how the emerging themes will be identified and analyzed.
Thaks you very much for devoting your time for reviewing the manuscript and giving us so important comments. In response to your suggestions we have improved greatly our manuscripts, explaining in section 2.9 “Method of conducting the qualitative” how data saturation will be assessed and how the emerging themes will be identified and analyzed.
-The discussion is concise and summarizes the study's findings with existing literature, identifies the significance of the research problem, and outlines practical implications for clinical practice and future research. However, to strengthen the discussion further, it could consider addressing potential limitations of the study and offering more concrete recommendations for translating research findings into actionable strategies for healthcare providers and policymakers.
Thank you very much for your valuable advice we have taken steps to address potential limitations of the study and offering more concrete recommendations for translating research findings into actionable strategies for healthcare providers and policymakers. This paragraph is reported in the discussions
-Furthermore, the connections between the study's findings and the broader context of cardiovascular health disparities should be highlighted so as to enhance the discussion's depth and relevance.
Thaks you very much for giving us so important comment. We appreciated and followed the suggested commentary, highlighting the connections between the study's findings and the broader context of cardiovascular health disparities should be highlighted so as to enhance the discussion's depth and relevance. This paragraph is reported in the discussions.
-This is a very good protocol. The abstract is well written with well-defined objectives, methodology is clear.
Thank you very much, we are glad you liked the protocol. We have been working on drafting this protocol since 2022.
-Please check and edit the grammar.
Thank you very much for the suggestion, we have arranged to have the English text revised by a native speaker expert.
-Under section 3.2 the group classification needs to be corrected to Groups I-III
Thank you for pointing out the confusion in our group classifications under section 3.2.. We have corrected the group labels to Groups I-III to ensure clarity.
-Potential limitations should be acknowledged.
Thank you for the opportunity to include possible limitations of this study in the discussions so that we can improve the protocol.
-How will the researchers address potential bias associated with qualitative sampling?
Thaks you very much for asking this! In the new version of the manuscript we have expanded this content as reported at the end of the discussions (“To avoid the potential bias associated with qualitative sampling, the criteria of Lincoln and Guba,1985 [54] will be used, which will ensure the scientific rigor of the present study. Continuous sampling up to data saturation ensured credibility. For the present study, credibility will be correspond to internal validity and will be the criterion for ensuring that the study examines what it was supposed to examine. Reliability will be guaranteed by the triangulation technique used in the study. The conduct of the data analysis by two or more researchers will allow confirmation the results from different perspectives. Confirmability will be guaranteed by the briefing of the participants and their confirmation of the themes extracted from the data obtained from them. Transferability will be guaranted through the participants’ in-depth descriptions of their experiences and the description of their sociodemographic characteristics [54]”),
Reviewer 2 Report
Comments and Suggestions for Authors
The manuscript “Women Acute Myocardial infarction - Identifying and Under-2 Standing the Gender Gap (WAMy-GAP): A Study Protocol” reports a very interesting topic, exploring the problems of women with acute myocardial infarction and preparing a new protocol.
However, it needs to be improved.
The abstract should be clarified, the 1st sentence should be split. Also, describe the abbreviations to understand the context better.
Why is there such a difference between men ans women? Do the comparison, regarding the diagnose, and all the factors involved.
Lines 46-60: there are a lot of repetitions with the lines 40-45, please simplify.
Line 79-81: very confusing.
Lines 102-104: please explain.
Please better explain the methods, simplify them as well as the disussion.
Comments on the Quality of English LanguageOverall the English must be improved, there are some grammar mistakes. Some sentences are too long, they should be split and simplified, since they are very confusing. Some examples are in lines 40-45, 180-187, 189-203, and so on. Xorrect throughout the manuscript please.
Author Response
The manuscript “Women Acute Myocardial infarction - Identifying and Under-2 Standing the Gender Gap (WAMy-GAP): A Study Protocol” reports a very interesting topic, exploring the problems of women with acute myocardial infarction and preparing a new protocol.
-However, it needs to be improved.
Thaks you very much for devoting your time for reviewing our manuscript and giving us so important comments. Thanks to your comments we have improved greatly our manuscripts.
-The abstract should be clarified, the 1st sentence should be split. Also, describe the abbreviations to understand the context better.
Thanks you. We followed the suggestions, dividing the sentence and describing the abbreviations so as to make the text more understandable.
-Why is there such a difference between men ans women? Do the comparison, regarding the diagnose, and all the factors involved.
Thank you very much, we have included the comparison regarding diagnosis and factors involved in gender difference within the introduction.
-Lines 46-60: there are a lot of repetitions with the lines 40-45, please simplify.
-Line 79-81: very confusing.
-Lines 102-104: please explain.
Thank you for your insightful feedback. In response, we have simplify lines 46-60and 45-50. Additionally, we have revised lines 79-81 for clarity and provided a detailed explanation for lines 102-104 to enhance understanding.
-Please better explain the methods, simplify them as well as the disussion.
Thank you very much for the suggestion, we have been simplifying the methods by explaining them better and expanding the study discussions.
Reviewer 3 Report
Comments and Suggestions for Authors
Cardiovascular diseases affect a large part of the population and investigations related to their prevention, diagnosis and treatment are of real interest.
The study aimed to validate the Italian McSweeney Acute and Prodromal Myocardial Infarct Symptom Survey (MAPMISS) scale.
Line 168. Surely it is Group III? I think the authors got the classification wrong.
I recommend the authors to summarize the eligibility criteria in a table because they are difficult to read.
The questionnaire of socio-demographic variables can be uploaded as an additional sheet.
Author Response
-Cardiovascular diseases affect a large part of the population and investigations related to their prevention, diagnosis and treatment are of real interest.
-The study aimed to validate the Italian McSweeney Acute and Prodromal Myocardial Infarct Symptom Survey (MAPMISS) scale.
Thaks you very much for devoting your time for reviewing our manuscript and giving us so important comments.
-Line 168. Surely it is Group III? I think the authors got the classification wrong.
Thank you for your insightful feedback. You are correct, and we appreciate your pointing out the error regarding Group III's classification. We have corrected this in the manuscript to ensure accuracy
-I recommend the authors to summarize the eligibility criteria in a table because they are difficult to read.
Thank you for your careful scrutiny. We have taken care as indicated to summarize the eligibility criteria within tables which we believe enhances clarity and reader comprehension to make it easier and smoother to read
-The questionnaire of socio-demographic variables can be uploaded as an additional sheet.
Thank you very much for your suggestion. We have, as suggested, attached the questionnaire of socio-demographic variables as an additional sheet.

Reviewer 4 Report
Comments and Suggestions for Authors
The manuscript, “Women Acute Myocardial Infarction—Identifying and Understanding the Gender Gap (WAMy-GAP): A Study Protocol,” evaluates prodromal symptoms and identifies risk behaviors in women with acute myocardial infarction (AMI). This is an interesting topic that contributes to knowledge in the area, but certain issues must be corrected.
Major revisions
1. In the abstract and introduction, the authors must mention the gap the manuscript will fill within current knowledge.
2. The authors must add results and a conclusion to the introduction.
3. The theoretical framework and objectives must be summarized in the introduction and not occupy a section for this. It must be noted that within a scientific manuscript, the theoretical framework is part of the introduction.
4. The authors must follow a scientific method, showing their methods and the results of the experiments, which indicate a conclusion. On the other hand, if the authors propose a protocol, it must be specified in the abstract and the introduction, mainly because the manuscript does not show the results that the protocol would yield.
5. The methodology considers the important points for the protocol, so I do not suggest anything for this part of the manuscript, and no control should be added.
6. It is important that the authors mention in their conclusion that this is a protocol proposal and that, therefore, the conclusions of their manuscript are probabilistic and not supported by real results.
7. The authors must make a figure or diagram representing the steps in section 3.
8. In figure legends, authors must describe the flowchart.
Minor revisions
1. Define all abbreviations to be presented in the text. For instance, AMI is not defined.
Comments on the Quality of English Languageno comments
Author Response
-The manuscript, “Women Acute Myocardial Infarction—Identifying and Understanding the Gender Gap (WAMy-GAP): A Study Protocol,” evaluates prodromal symptoms and identifies risk behaviors in women with acute myocardial infarction (AMI). This is an interesting topic that contributes to knowledge in the area, but certain issues must be corrected.
Thaks you very much for devoting your time for reviewing our manuscript and giving us so important comments.
Major revisions
- In the abstract and introduction, the authors must mention the gap the manuscript will fill within current knowledge.
Thank you very much for the suggestion, we have been very pleased with the inclusion of the gaps that the study aims to fill.
- The authors must add results and a conclusion to the introduction.
Thank you very much for the comment, we have added the conclusion; since this manuscript describes a study protocol, it does not include results; these will be presented upon completion of the research. We appreciate your guidance and have adjusted the manuscript accordingly.
- The theoretical framework and objectives must be summarized in the introduction and not occupy a section for this. It must be noted that within a scientific manuscript, the theoretical framework is part of the introduction.
Thank you, we made the requested corrections, which were very helpful and improved the layout of our manuscript.
- The authors must follow a scientific method, showing their methods and the results of the experiments, which indicate a conclusion. On the other hand, if the authors propose a protocol, it must be specified in the abstract and the introduction, mainly because the manuscript does not show the results that the protocol would yield.
Thank you very much for the suggestion, we have included in the abstract and introduction that this is a protocol study so the results are not present.
- The methodology considers the important points for the protocol, so I do not suggest anything for this part of the manuscript, and no control should be added.
Thank you very much.
- It is important that the authors mention in their conclusion that this is a protocol proposal and that, therefore, the conclusions of their manuscript are probabilistic and not supported by real results.
Thank you very much for the suggestion. we agree it is necessary to include that it is a protocol.
- The authors must make a figure or diagram representing the steps in section 3.
Thank you very much for the suggestion, we have made a diagram representing the steps in section 3, so that the process can be better understood.
- In figure legends, authors must describe the flowchart.
Thank you for your careful scrutiny, we have taken care of the flowchart description
Minor revisions
- Define all abbreviations to be presented in the text. For instance, AMI is not defined.
Thank you very much for the attention brought back to the manuscript, We have proceeded with the changes.
Round 2
Reviewer 2 Report
Comments and Suggestions for Authors
Overall, the authors made a great effort since the manuscript was improved. However, the English must be improved, there are some grammar mistakes.
Some examples:
Lines 162 and 184: replace "simple" with "sample".
Lines 202 and 208: It should be "The qualitative/quantitative study will be sampled according to the following criteria"
Comments on the Quality of English Language
The English must be greatly improved, there are some grammar mistakes.
Some examples:
Lines 162 and 184: replace "simple" with "sample".
Lines 202 and 208: It should be "The qualitative/quantitative study will be sampled according to the following criteria"
Author Response
Overall, the authors made a great effort since the manuscript was improved. However, the English must be improved, there are some grammar mistakes.Some examples: Lines 162 and 184: replace "simple" with "sample".
Lines 202 and 208: It should be "The qualitative/quantitative study will be sampled according to the following criteria"
Thank you very much for taking the time to review the study protocol and linguistic advice. We have made the requested modifications, performed a second linguistic editing, and improved the linguistic form
Comments on the Quality of English Language
The English must be greatly improved, there are some grammar mistakes.
Some examples: Lines 162 and 184: replace "simple" with "sample".
Lines 202 and 208: It should be "The qualitative/quantitative study will be sampled according to the following criteria"
Thank you very much for taking the time to review the study protocol and linguistic advice. We have made the requested modifications, performed a second linguistic editing, and improved the linguistic form. The changes are indicated in blue color.
